# OpenReview forum: "TUBench: Benchmarking Large Vision-Language Models on Trustworthiness with Unanswerable Questions"
_ICLR.cc/2025/Conference — Submitted to ICLR 2025_

### Official Review · Reviewer_653o · 2024-10-31

**Soundness:** 3
**Presentation:** 3
**Contribution:** 3
**Rating:** 6
**Confidence:** 4

**Summary:**

This paper introduces a new benchmark called TUBench designed to evaluate Large Vision-Language Models (LVLMs) for reliability using unanswerable questions. Unlike previous benchmarks that focus on answerable visual questions, TUBench uses unanswerable queries based on images from diverse domains like code snippets, natural scenes, geometry, and statistical tables to test the models' trustworthiness. Through experiments on 28 models, the study reveals that hallucination is a primary challenge, with even top models like Gemini-1.5-Pro achieving only modest success in distinguishing answerability, highlighting significant gaps in current LVLM capabilities.

**Strengths:**

1. This paper focuses on unanswerable questions across diverse image types (e.g., code snippets, natural scenes, geometric diagrams, and statistical tables), TUBench provides a unique angle for assessing LVLM trustworthiness, setting it apart from benchmarks that focus only on answerable questions.
2. TUBench’s dataset of 2,354 questions includes 1,151 unanswerable questions constructed using 10 strategies. The manual curation ensures high quality and relevance, providing nuanced and challenging unanswerable queries.
3.The paper includes detailed analyses of model errors, specifically investigating the role of hallucination and overconfidence in LVLMs when faced with unanswerable questions. This analysis underscores a critical shortcoming in current models.

**Weaknesses:**

1. Can the authors provide suggestions for future LVLM training to solve the "unanswerable" questions? From both the model perspective and training data perspective.

2. Although human-annotated questions ensure high quality, manual creation of unanswerable questions may introduce subjective biases or inconsistencies, which could affect reproducibility or lead to unintentional patterns that models might exploit.

**Questions:**

The questions are mentioned in the weakness section.

---

> ### Author Response · Authors · 2024-11-24
> **Response to Reviewer 653o**
>
> **Weakness1:** Can the authors provide suggestions for future LVLM training to solve the "unanswerable" questions? From both the model perspective and training data perspective.
>
> Thank you for your question. The following methods could enhance LVLMs’ performance on unanswerable questions from both model and training data perspectives:
>
> 1. Model Perspective:  LVLMs may benefit from mechanisms to explicitly model uncertainty and determine when they lack the information needed to answer a question. For instance, training models with objectives focused on uncertainty estimation, such as confidence calibration or abstention techniques, could enable them to recognize when a question is unanswerable.
>
> 2. Training Data Perspective: A likely reason for LVLMs’ lower performance on TUBench’s unanswerable questions is their limited exposure to unanswerable scenarios during instruction tuning, which primarily features answerable examples. To address this,
> (a) Expanding Unanswerable Training Samples: Developing a large, diverse set of unanswerable training questions would expose models to various unanswerable conditions, allowing them to better identify missing information across different contexts. (b) Targeted Data Augmentation: Crafting datasets where answerable and unanswerable questions are generated from similar images could help the model learn subtle cues that distinguish answerable from unanswerable questions.
>
> **Weakness2:** Although human-annotated questions ensure high quality, manual creation of unanswerable questions may introduce subjective biases or inconsistencies, which could affect reproducibility or lead to unintentional patterns that models might exploit.
>
> We appreciate the insightful comment regarding the potential for subjective biases or inconsistencies in manually crafted unanswerable questions. However, to mitigate these concerns, we have considered the following aspects during the construction of the TUBench dataset:
> 1. Diverse Question Creation Strategies: Unanswerable questions in TUBench were generated using ten distinct strategies tailored to different contexts (e.g., occlusion, unclear details, and uncertain spatial relationships). These strategies ensured a systematic approach that minimized reliance on individual annotator subjectivity (see Section 3.1 of the paper for details).
> 2. Diverse Visual Contexts: TUBench spans multiple domains (e.g., code reasoning, natural images, geometric diagrams, statistical tables), further reducing the risk of domain-specific biases or reproducibility issues.
> 3. Quality Control: Each manually created question was reviewed by three annotators to assess its relevance, accuracy, and adherence to the guidelines. Questions flagged by more than half of the reviewers were excluded from the final dataset. The inter-annotator agreement, measured by Fleiss’ kappa, is 0.71, indicating substantial agreement among annotators (greater than 0.6) (more details in Section B on page 18). This rigorous validation process reduces the likelihood of subjective bias influencing the quality and reproducibility of the dataset.
>
> As demonstrated in Table 2 on page7, even the best-performing models struggled to identify unanswerable questions, indicating that TUBench does not contain patterns easily exploitable by models. While no dataset is entirely free from bias, our systematic approach and rigorous quality control procedures aim to ensure that TUBench provides a robust and reproducible evaluation framework. We welcome suggestions for further enhancing its quality.

---

> ### Author Response · Authors · 2024-12-02
> **Follow-up on Our Response to Your Review Comments**
>
> Dear reviewer 653o,
>
> We would like to kindly follow up on our previous response to your comments and hope that the clarifications provided have addressed your concerns. As the discussion period is nearing its end, if there are any remaining questions or issues with our response, we would appreciate it if you could let us know as soon as possible. We are committed to resolving any remaining concerns to your satisfaction.
> In addition, in response to the reviewers' valuable feedback, we have updated the manuscript and added new experiments. In light of these efforts and the additional findings we have presented, we would kindly ask you to reconsider your evaluation.
>
> Thank you again for your time and thoughtful consideration.

---

### Official Review · Reviewer_gyxQ · 2024-11-02

**Soundness:** 2
**Presentation:** 3
**Contribution:** 3
**Rating:** 5
**Confidence:** 4

**Summary:**

The paper introduces TUBench, a novel benchmark specifically designed to assess the reliability of large vision-language models (LVLMs) when faced with unanswerable questions. TUBench includes a diverse set of high-quality, manually crafted unanswerable questions across four visual domains. This benchmark evaluates trustworthiness by focusing on LVLMs' ability to refuse to answer when information is insufficient. The contributions are:
1. A systematic dataset of 2,354 questions split into answerable and unanswerable subsets, carefully crafted to challenge LVLMs.
2. A comprehensive evaluation of 28 LVLMs, revealing substantial room for improvement in identifying unanswerable questions.
3. Analysis showing hallucination as a primary source of errors, highlighting the gap in current LVLMs' reliability in real-world applications.

**Strengths:**

1. TUBench introduces a new problem and a new dataset for evaluating Large Vision-Language Models (LVLMs) by focusing on unanswerable questions.
2. The TUBench is well-structured with various domains that can happen in real life. The evaluation scale is large, containing 28 MLLMs.
3. The paper is mostly well-written and easy to follow.

**Weaknesses:**

1. Some examples may be a bit not rigorous, for instance, in the bottom left subfigure in Fig 1, if the reason that the code snippet is not complete leads to the unanswerable for the first two questions, then the last two questions are also not answerable: other part of the code may change the result.
2. The original performance of MLLMs on the GeoQA dataset seems very low; in this case, it does not make much sense to expect them to judge the answerability of the question by removing some conditions. Mostly, one has to know how to solve the problem to judge the answerability of it. About this point, can the authors show the best results of MLLMs on the answerable subset of the dataset?

**Questions:**

1. Line 414 refers to Figure 13, but Figure 13 does not show that GPT4o is wrong in question answerability; it just got the wrong answer. This example seems marginal.
2. The paper could address more on the significance/importance of the proposed problem: what bad outcomes in which situations would the weakness of MLLMs on this problem lead?
3. Does the 'hallucination' mentioned in the paper mean 'object hallucination'? How is the error in the UCR dataset classified in The result in Fig 5 (b)?
4. It is not quite clear how exactly OAAC is calculated, can the author provide the equation?

---

> ### Author Response · Authors · 2024-11-24
> **Response to Reviewer gyxQ [1/2]**
>
> **Weakness1:**  Some examples may be a bit not rigorous.
>
> Thank you for your thoughtful comments. We appreciate your attention to detail regarding the example in the bottom left subfigure of Figure 1. Here is our clarification:
>
> * Scope of Code Context: During our evaluation, we require models to base their answers strictly on the code visible in the provided image. The assumption of additional, unseen code beyond the displayed snippet exceeds the scope of the information given. Notably, the code in the figure does not imply further content, as there are no ellipses (...) or comments indicating continuation. Assuming the existence of hidden code introduces unsupported conjecture.
>
> * Reasonable Inference from Shown Code: For the existing portion of the code (e.g., where variable x is uninitialized), it is a logical and contextually valid inference to deduce the outputs depend on the value of x. While the complete output is uncertain, certain outcomes (e.g., "hello") can be reasonably ruled out based on the visible information. Thus, our methodology reflects logical inference based on provided information rather than speculation.
>
> * Purpose of the Design: The goal of such examples is to test whether models can accurately determine when a question is unanswerable based on visible information, without making assumptions about non-existent elements. By carefully curating the code snippet and its context, we aim to assess the model's ability to discern the sufficiency of available data to answer questions.
>
> We hope this explanation addresses your concerns, and we welcome any further discussion or suggestions!
>
>
> **Weakness2:** About this point, can the authors show the best results of MLLMs on the answerable subset of the dataset?
>
> Thank you for your insightful comment. We have addressed this concern by presenting the evaluation results for both the answerable and unanswerable subsets of TuBench in Table 11 of the updated manuscript. Please refer to Table 11 and the accompanying analysis on Page 22 for detailed insights.
>
> As shown in Table 11, Gemini-1.5-Pro achieved the highest OACC score of 65.7\% on the answerable subset, which is significantly better than the random guess baseline of 20\%. However, it performed poorly on the unanswerable subset, achieving an accuracy of only 26.7\%, which is far below the random guess baseline of 50\%. In contrast, Gemini-1.5-Flash achieved an OACC score of 36.3\% on the answerable subset but demonstrated superior performance on the unanswerable subset with a 2ACC of 52.2\%, substantially outperforming Gemini-1.5-Pro in this aspect.
>
> These results highlight an important observation: the ability of VLMs to solve answerable questions does not always align with their ability to determine whether a question is answerable. We believe this emphasizes the need for further exploration and improvement of VLMs in handling unanswerable questions effectively.
>
>
> **Q1:** Line 414 refers to Figure 13, but Figure 13 does not show that GPT4o is wrong in question answerability; it just got the wrong answer. This example seems marginal.
>
> Thank you for your suggestion. We have carefully reviewed the example referred to in Figure 13 and agree that it does not clearly demonstrate that GPT-4o was wrong in determining question answerability. To address this, we have replaced the example with two new examples where hallucinated content led the models to incorrectly assess the answerability of the questions. Please refer to Line 414,  Figure 14, and Figure 15 in the updated manuscript for these new examples.
>
> **Q2:**  What bad outcomes in which situations would the weakness of MLLMs on this problem lead?
>
> Thank you for the insightful comment. The significance of the proposed problem lies in addressing the critical risks posed by hallucinations of VLMs when encountering unanswerable questions. Prior research has shown that these models may generate confident yet incorrect responses even when the necessary information is absent (Yin et al., 2023b; Sun et al., 2024). This behavior is particularly concerning in domains requiring high reliability, such as medical diagnosis and autonomous driving. For example, in medical contexts, a hallucinated diagnosis or treatment recommendation could lead to severe health consequences or even loss of life. Similarly, in autonomous driving, an incorrect response to a question about environmental conditions or vehicle status could result in unsafe decisions, potentially causing accidents.
>
> The expectation for models in these situations is clear: they should refrain from generating fabricated responses and instead explicitly acknowledge their inability to answer. Addressing this issue is essential to ensure the trustworthy deployment of MLLMs in critical applications, highlighting the importance of systematically evaluating and mitigating hallucination risks.
>
> We highlight the research significance of our research in Lines 57-64 of the paper.

---

> > ### Comment · Reviewer_gyxQ · 2024-11-27
> > **Thank you for the reply**
> >
> > **W1** It is still unclear to me whether there is a coherent standard for answerability in UCR. If the assumption of no hidden/unseen code is not allowed, then the answers to the bottom right example in Figure 1 should be changed because the code has a bug, and no output will be generated for sure.
> >
> > **W2** Thanks to the authors for the extra experiments. However, the results in Table 11 seem to lead to some more questions. Firstly, it does show that most of the models in UGeoQA and UCR have very low performance (on UCR it seems the performances are similar to random), which is potentially one of the main reasons why the 2ACC is low. Second, on the above two datasets, there are cases when the overall 2ACC between models is similar, but the 2ACC on individual subsets is very different. I feel like these (and potentially more) observations need more explanations to make the behaviors of the models clearer.

---

> ### Author Response · Authors · 2024-11-24
> **Response to Reviewer gyxQ [2/2]**
>
> **Q3:** Does the 'hallucination' mentioned in the paper mean 'object hallucination'? How is the error in the UCR dataset classified in The result in Fig 5 (b)?
>
> Thank you for your question. As stated in Line 426, the term 'hallucination' in this paper refers to content that is inconsistent with the image, and it is not limited to `object hallucination'. We provide detailed examples in Figures 13–16 in Section D.2 (page 26) of the updated manuscript. Specifically:
> * Figures 13 and 14 illustrate cases of object hallucination, where the generated content includes objects that are not present in the image.
> * Figures 15 and 16 depict instances where the generated content is inconsistent with the information in the image, which extends beyond just object hallucination.
>
> Regarding the errors in the UCR dataset presented in Figure 5(b), these errors are further analyzed in Figures 15, 17, and 19 in the updated manuscript:
> Figure 15 showcases examples of hallucinations in the generated content.
> Figure 17 provides an example of errors caused by wrong calculations in the explanations generated by VLMs.
> Figure 19 demonstrates errors that fall outside of hallucinations or wrong calculation. For more detailed examples and analysis, please refer to Section D.2 (page 26), Section D.3 (page 30), and Section D.4 (page 32) in the updated manuscript.
>
>
> **Q4:** Can the author provide the equation.
>
> The equation for OACC is:
>
> $\text{OACC} = \frac{C_u + C_a}{Q_u + Q_a}$
>
> Where:
>
>    *  $Q_u$ is the total number of questions in the unanswerable set.
>    *  $Q_a$ is the total number of questions in the answerable set.
>    *  $C_u$ is the number of unanswerable questions correctly identified as unanswerable.
>    *  $C_a$ is the number of answerable questions for which the model correctly determines that the questions are answerable and provides the correct answers.

---

> ### Author Response · Authors · 2024-12-02
> **Response to Reviewer gyxQ [1/3]**
>
> Thank you for your follow-up comments.
>
> **1 Response to** " It is still unclear to me whether there is a coherent standard for answerability in UCR. If the assumption of no hidden/unseen code is not allowed, then the answers to the bottom right example in Figure 1 should be changed because the code has a bug, and no output will be generated for sure."
>
> Our methodology adheres to a consistent standard: models are required to base their reasoning strictly on the information provided in the images, avoiding unsupported assumptions or excessive reasoning. For the code reasoning dataset, models should derive their conclusions from the code snippets provided in the screenshots. The reviewer’s concern about "other parts of the code may change the result" represents an unsupported and excessive inference.
>
>
> First, there is no evidence in the code snippets (e.g., the bottom-left and bottom-right of Figure 1) to suggest that additional code segments exist in the original code that could alter the output. Thus, this reasoning lacks a foundation and constitutes excessive speculation. Adopting such a perspective would render all questions related to the code snippet in top-left of Figure 1 unanswerable. Moreover, other widely used code reasoning datasets, such as MME [1], would face analogous challenges if we accepted this excessive reasoning standard.
>
>
> When constructing the code reasoning dataset, all of our reasoning is based strictly on the code provided in the screenshots. For example, in the bottom-left snippet of Figure 1, the output depends on the value of the variable x. Since the image does not provide a value for x, the output of the code depends entirely on its value. However, there is no evidence in the image suggesting the presence of additional code lines that could modify the result. Thus, we can conclude that the output of the code can only be “Hello” or “World” and cannot be “hello” or “world.”
>
> Similarly, for the bottom-right code snippet in Figure 1, we can infer that the output involves an incomplete segment related to two print statements. While the specific output cannot be determined due to the incompleted code lines, there is no evidence in the snippet to support the assumption that other code segments outside the image might modify the output. Thus, the reasonable conclusion is that the original output begins with “Hello,” based on the visible information.
>
> [1] MME: A Comprehensive Evaluation Benchmark for Multimodal Large Language Models (https://arxiv.org/abs/2306.13394)

---

> > ### Author Response · Authors · 2024-12-02
> > **Response to Reviewer gyxQ [2/3]**
> >
> > **2 Response to** ''it does show that most of the models in UGeoQA and UCR have very low performance (on UCR it seems the performances are similar to random), which is potentially one of the main reasons why the 2ACC is low.''
> >
> > The OACC score of all models on UGeoQA's answerable set is higher than the OACC score of random guessing (i.e., 20\%). However, in UCR, the OACC score of GPT-4 models on the answerable set is lower than the random guessing baseline (i.e., 33.3\%). The low OACC scores of GPT-4 models in UCR are mainly due to their frequent inability to identify the questions in the answerable set are indeed answerable. For example, GPT-4 Turbo can only correctly recognize 51.5\% of the answerable questions, failing to identify 48.5\% of the answerable questions. For those questions it does recognize as answerable, models should further provide the correct answer in order to improve the OACC score. Therefore, we cannot attribute the low 2ACC in UCR to the low OACC score. In fact, from the above analysis, the OACC score on the answerable set is closely related to the 2ACC score on the answerable set.
> >
> > However, the ability of VLMs to correctly answer questions within the answerable set (i.e., OACC on the answerable set) does not always correlate with their ability to identify unanswerable questions in the unanswerable set (i.e., 2ACC on the unanswerable set). This is evident from the OACC results displayed in Table 11 for the answerable subsets. The OACC metric assesses not only whether VLMs can determine if a question is answerable, but also whether they can provide the correct answer when it is answerable. While the average OACC scores of Qwen-VL models are only slightly lower than those of the best-performing models, such as Gemini-1.5-Pro and GPT-4, this indicates that Qwen-VL models do have some ability to solve answerable questions. However, the significantly lower 2ACC results of Qwen-VL models on the unanswerable subsets show poor performance in distinguishing unanswerable questions. Furthermore, Qwen-VL models achieve much higher OACC scores (47.4\% and 44.4\%) on the answerable subset of UCR compared to the random baseline (33.3\%), but their 2ACC scores on the unanswerable subset are very low (9.3\% and 15.4\%). This suggests that Qwen-VL models tend to classify questions as answerable even when they are not, indicating an overconfidence in distinguishing question answerability. Therefore, while Qwen-VL models show some competence in solving answerable questions, their performance in identifying unanswerable ones is notably weak.
> >
> > In summary, the low 2ACC scores of VLMs on UCR and UGeoQA highlight the significant challenge our datasets pose to existing models, underscoring the urgent need to improve the ability of current models to handle unanswerable questions.

---

> ### Author Response · Authors · 2024-12-02
> **Response to Reviewer gyxQ [3/3]**
>
> **3 Response to** ''Second, on the above two datasets, there are cases when the overall 2ACC between models is similar, but the 2ACC on individual subsets is very different. I feel like these (and potentially more) observations need more explanations to make the behaviors of the models clearer.''
>
> Upon comparing Table 2 and Table 11, we observe that while the overall 2ACC scores of some models are similar, the 2ACC scores on the unanswerable and answerable sets, show noticeable differences. This suggests that the models exhibit different inherent tendencies. For instance, Qwen-VL models tend to classify questions as answerable, even when they are actually unanswerable. In contrast, Gemini and GPT-4 models have lower inherent tendencies to classify questions as answerable, indicated by the lower average 2ACC scores on the answerable set, but higher 2ACC scores on the unanswerable set.
>
> We hope our explanation addresses your concerns, and we welcome any further discussion or suggestions!

---

### Official Review · Reviewer_KY3C · 2024-11-04

**Soundness:** 2
**Presentation:** 3
**Contribution:** 2
**Rating:** 5
**Confidence:** 3

**Summary:**

This paper presents a new benchmark to evaluate the reliability of LVLMs with both answerable and unanswerable questions, which is different from the SOTA ones that focus on answerable questions. The benchmark contains images from four domains including code snippet screen shots, natural images, geometry diagrams and statistical table screen shots. The benchmark is used to evaluate 28 foundational models in terms of determining question's answerability and others. The evaluation results indicate that the STOA LVLMs do not perform well in answering unanswerable questions.

**Strengths:**

1. The presented work is well motivated. While the existing benchmarks overlook unanswerable questions or used heuristic rules to automatically generated unanswerable questions, this proposed new benchmark that create unanswerable yet closely related questions and present a mixture of answerable and unanswerable ones to evaluate LVLMs.

2. The evaluation of 28 leading foundation LVLMs using the new benchmark reveals low accuracy even for the best-performing model, which indicates the need for improvement.

3. The paper is well organized and written, with clear and detailed description of the benchmark, the evaluation methodology and evaluation results.

**Weaknesses:**

1. It would be better to study and compare the performance of the LVLM models when they are presented with only answerable questions and with a mixture of answerable and unanswerable questions. Such a ablation study would help to quantify the impact of the presence of unanswerable questions. The current study only uses a mixture of answerable/unanswerable questions.

2. It is claimed that identifying answerability is more challenging, but this claim needs more clear/systematic justification. For example, on page 9 line 450 - "unanswerable questions are challenging for current VLMs, with out 44% of Gemini-1.5-pro's outputs providing correct answers and explanations"; however, according to Fig 5 (a), it appears that for both answerable and unanswerable questions, only 41% of Gemini-1.5-Pro's outputs provide correct answers and explanations. Does this indicate correctly answering the answerable questions is also challenging?

**Questions:**

Among the evaluation metrics, F1 and 2ACC are already used to measure the performance of identifying unanswerable questions, why is OACC introduced to measure the combination of both accuracy of answerability classification and the accuracy for answerable questions? Is the redundant? Is it better to have the third metric measure only the accuracy of answerable questions?

---

> ### Author Response · Authors · 2024-11-24
> **Response to Reviewer KY3C**
>
> **Weakness1:** Present the performance of the LVLM models with  answerable questions and unanswerable questions.
>
> Thank you for your suggestion. We present the evaluation results for the unanswerable and answerable subsets of TUBench in Table 11 of the updated manuscript. For more details, we invite you to refer to Table 11 and the corresponding analysis provided on Page 22 of the revised manuscript.
>
>
> **Weakness2:** Does this indicate correctly answering the answerable questions is also challenging?
>
> Figure 5(a) presents the human evaluation results for 50 answerable and 50 unanswerable questions—not just answerable questions. Figures 15(a) and 16(a) show the results for the answerable and unanswerable subsets, respectively. We speculate that the reviewer is asking whether, based on Figure 15(a), it can be inferred that correctly answering the answerable questions is also challenging.
>
> It is important to note that although the results for answerable questions are shown in Figure 15(a), the models did not know in advance whether the questions are answerable. To correctly answer these questions, the models must first determine that the questions are indeed answerable and then provide the correct answers (as measured by the OACC metric).
>
> From Figure 15(a), we can infer that it is challenging for the models to both determine the answerability of questions in the answerable subset and correctly answer them. However, this does not imply that correctly answering questions in the answerable subset is challenging, since this conclusion assumes that the models already know the questions are answerable and only need to provide the correct answers.
>
> The reviewer may also refer to our response to \#Q1\# below for a clearer understanding of the distinction between these two aspects.
>
>
>
> **Q1:** Why is OACC introduced to measure the combination of both accuracy of answerability classification and the accuracy for answerable questions? Is the redundant?
>
> Thank you for your thoughtful question. The primary goal of our paper is to evaluate how models perform when faced with unanswerable questions. Therefore, our fundamental assumption is that the answerability of a question is unknown beforehand.
>
> Introducing a metric that focuses solely on the accuracy of answerable questions would require informing the models in advance whether the questions are answerable. This assumption contradicts those of the first two evaluation metrics, F1 and 2ACC, which evaluate performance without prior knowledge of answerability. Moreover, due to this inconsistency in assumptions, we would need to re-evaluate the models under the new assumption for the same set of questions. For the GPT-4 series models, this would incur additional evaluation costs.
>
> In contrast, the OACC metric provides a holistic view by simultaneously considering the accuracy of answerability classification and the accuracy for answerable questions. Importantly, it adheres to the same assumption as the F1 and 2ACC metrics, enabling us to evaluate all three metrics in a single evaluation run. This consistency streamlines the evaluation process and avoids additional computational and financial overhead.
>
> We hope this clarification demonstrates why OACC is not redundant but instead complements the other metrics by maintaining consistency while offering a broader perspective. For the prompts used in our evaluation, please refer to Table 7 on page 20 of the revised paper.

---

> > ### Author Response · Authors · 2024-12-02
> > **Follow-up on Our Response to Your Review Comments**
> >
> > Dear reviewer KY3C,
> >
> > We would like to kindly follow up on our previous response to your comments and hope that the clarifications provided have addressed your concerns. As the discussion period is nearing its end, if there are any remaining questions or issues with our response, we would appreciate it if you could let us know as soon as possible. We are committed to resolving any remaining concerns to your satisfaction. In addition, in response to the reviewers' valuable feedback, we have updated the manuscript and added new experiments. In light of these efforts and the additional findings we have presented, we would kindly ask you to reconsider your evaluation.
> >
> > Thank you again for your time and thoughtful consideration.

---

### Official Review · Reviewer_m7Wd · 2024-11-04

**Soundness:** 2
**Presentation:** 2
**Contribution:** 2
**Rating:** 5
**Confidence:** 4

**Summary:**

This paper introduces TUBench, a benchmark aimed at assessing the reliability of Language-Vision Models (LVLMs) by using unanswerable questions. TUBench features a diverse set of high-quality unanswerable questions, created using ten distinct strategies, to rigorously test LVLMs’ responses. The benchmark draws from four varied visual domains—code snippets, natural images, geometry diagrams, and statistical tables—to examine trustworthiness in specific reasoning areas: code, commonsense, geometry, and mathematical reasoning.

**Strengths:**

1. This paper evaluates the trustworthiness of VLMs across four reasoning domains applying 10 approaches to address unanswerability.
2. The paper provides in-depth analysis of model performance, including sensitivity to hallucination and question answerability, which gives a nuanced understanding of model limitations.

**Weaknesses:**

1. The generation of approach for UVQA and UTabMWP domains are less diverse; specifically in UVQA, perturbations in relations/attributes using scene graphs to generate unanswerable questions have been already addressed in current literature [a].  The fixed number of perturbations can easily be included in a few-shot way to the VLM to teach unanswerability to VLM.
2. The effect of changing the ordering of MCQ options is not explored, although prior works have found bias in VLMs in option ordering.

(a) https://arxiv.org/pdf/2403.10534

**Questions:**

Comment: Recent works tried to address unanswerability [a] by perturbing relations/attributes and hallucinations [b] of VLMs which are worth exploring.

Q1. UCR, UVQA are posed in Y, N, U option order. Has there been analysis with changing the option order?

Q2. In UTabMWP, table being incomplete derives in Unanswerable situation which is very easy to embed into the VLM with few-shot examples.

Q3. UGeoQA contains questions in Chinese and most VLMs are predominantly English. Therefore, it is unclear if the ineffectiveness of the VLMs derive from inability to process Chinese or actual reasoning.

Q4. What the 'Other' category of explanation evaluation categories consists?




(a) https://arxiv.org/pdf/2403.10534
(b) https://www.researchgate.net/publication/375595895_HALLUSIONBENCH_You_See_What_You_Think_Or_You_Think_What_You_See_An_Image-Context_Reasoning_Benchmark_Challenging_for_GPT-4Vision_LLaVA-15_and_Other_Multi-modality_Models

---

> ### Author Response · Authors · 2024-11-24
> **Response to Reviewer m7Wd [1/2]**
>
> **Weaknesses1:** The generation of approach for UVQA and UTabMWP domains are less diverse; specifically in UVQA, perturbations in relations/attributes using scene graphs to generate unanswerable questions have been already addressed in current literature [a].
>
> Thank you for your insightful comment! The primary goal of our work is to evaluate the performance of VLMs when dealing with unanswerable questions. Literature [a] primarily generates unanswerable questions using two strategies: (1) objects mentioned in the question are absent in the image, and (2) the objects exist, but those with specific attributes are missing. In contrast, our proposed TUBench adopts ten diverse strategies to construct unanswerable questions.
>
> For the UVQA dataset, we use strategies such as occlusion of objects in the image (S.4), unclear object details (S.5), missing overall object information (S.6), and uncertain spatial relationships (S.7) to construct unanswerable questions. These strategies provide a broader diversity of unanswerable scenarios compared to literature [a]. Furthermore, while literature [a] uses natural images exclusively, TUBench integrates visual contexts from four distinct domains: natural images, code snippets, geometry diagrams, and statistical tables, adding significant diversity to the data.
>
> We also compared our unanswerable question construction approach with the image replacement method (detailed on page 10, ''Comparison to Image Replacement in UVQA''). Image replacement typically generates unanswerable questions by substituting original images with other ones, where unanswerability arises because the objects mentioned in the questions are absent from the images. Our results, supported by examples in Figures 24, 25, and 26 (page 36), indicate that questions constructed with unclear details, and spatial uncertainties pose a greater challenge to models.
> Specifically, Figure 24 highlights unanswerable questions caused by uncertain spatial relationships and unclear details. In contrast, Figures 25 and 26 feature images where the objects mentioned in the questions are entirely absent. Figure 25 is semantically similar to Figure 24, whereas Figure 26 is semantically distant from it. We observed that VLMs found it relatively easy to recognize the unanswerability of questions in Figures 25 and 26 but struggled significantly more with those in Figure 24.
>
> In summary, TUBench presents a significantly more challenging and diverse benchmark for evaluating VLMs' trustworthiness compared to prior works. For further details and comparisons, please refer to the examples and experimental results in Sections 4.4 (page 10) and D.7 (page 36) of our updated paper.
>
>
> **Weaknesses2 \& Q1:** UCR, UVQA are posed in Y, N, U option order. Has there been analysis with changing the option order?
>
> Thank you for your insightful comment! We evaluated the impact of option order on VLMs including GPT-4o on the UCR and UVQA datasets. Our analysis shows that while adjusting the option order has a slight impact on the performance of VLMs, it does not improve their ability to determine whether a question is answerable. This highlights the significant challenges posed by the proposed datasets to current VLMs. For further details, please refer to Section D.8 on page 39 of our paper.
>
> **Q2:** In UTabMWP, table being incomplete derives in Unanswerable situation which is very easy to embed into the VLM with few-shot examples.
>
> Thank you for your insightful comment! We conducted experiments with VLMs including GPT-4o under 3-shot and 6-shot settings. However, the ability of GPT-4o to determine whether a question is answerable did not show significant improvement in the few-shot setting. This indicates that simply providing demonstration examples is insufficient to address the issue effectively. For further details, we kindly refer you to Section D.9 on page 39 of our updated paper.

---

> ### Author Response · Authors · 2024-11-24
> **Response to Reviewer m7Wd [2/2]**
>
> **Q3:** UGeoQA contains questions in Chinese and most VLMs are predominantly English. Therefore, it is unclear if the ineffectiveness of the VLMs derive from inability to process Chinese or actual reasoning.
>
> Thank you for your insightful question. It is widely recognized that Chinese is not a low-resource language, and all seven proprietary models evaluated in our study support multiple languages, including Chinese. Therefore, the poor performance of these models on UGeoQA is primarily due to their challenges in handling unanswerable questions rather than an inability to process the Chinese language.
>
> To further clarify, we present a detailed comparison of model performance on answerable and unanswerable subsets of TUBench, including UGeoQA, in Table 11 on page 22.  We invite you to refer to the updated manuscript. The results demonstrate that proprietary VLMs generally perform significantly better on answerable subsets than on unanswerable subsets. For example, Gemini-1.5-Pro achieved a 2ACC score of 95.9\% on the answerable subset. However, Gemini-1.5-Pro performed poorly on the unanswerable subset, achieving a 2ACC score of only 26.7\%, which is substantially below the random guess baseline of 50\%. Moreover, Gemini-1.5-Pro achieved an OACC score of 65.7\% on the answerable subset, far exceeding the random prediction baseline of 20\%. This indicates that it possesses a certain level of competence in addressing answerable questions of UGeoQA. This stark contrast underscores that the ineffectiveness of these models on UGeoQA stems primarily from their inability to handle unanswerable questions effectively, and is not due to the models' limitations in processing the Chinese language.
>
> **Q4:** What the 'Other' category of explanation evaluation categories consists?
>
> The 'Other' category in the explanation evaluation refers to errors that do not fall under the categories of hallucinations or wrong calculations. We provide two illustrative examples of other errors in Section D.4 on pages 32-33 of the updated manuscript. In the first example, the error occurs because the VLMs overlooks the case-sensitive nature of the output required by the question. In the second example, the error arises from the model's wrong conclusion.
> For more details, we invite you to refer to Section D.4 on pages 32-33 of the revised manuscript.

---

> ### Author Response · Authors · 2024-12-02
> **Follow-up on Our Response to Your Review Comments**
>
> Dear reviewer m7Wd,
>
> We would like to kindly follow up on our previous response to your comments and hope that the clarifications provided have addressed your concerns. As the discussion period is nearing its end, if there are any remaining questions or issues with our response, we would appreciate it if you could let us know as soon as possible. We are committed to resolving any remaining concerns to your satisfaction. In addition, in response to the reviewers' valuable feedback, we have updated the manuscript and added new experiments. In light of these efforts and the additional findings we have presented, we would kindly ask you to reconsider your evaluation.
>
> Thank you again for your time and thoughtful consideration.

---

### Meta-Review · Area_Chair_ZiF5 · 2024-12-22

**Metareview:**

This work presentend a new benchmarks called TUBench, aimed at examining the trustworthness of VLMs. The authors pointed that LLMs and VLMs incline to produce answers to questions, even for those that are not answerable given the input image and text. To evaluate the VLMs more comprehensively, the authors developed a new QA datasets which explicitly contains both answerable and unanswerable questions for given images. The authors deliberatly curated the image QA pairs from four different domains and perform some variantions to the original QA pairs to make them unanswerable. Based on this new datasets, the authors conducted extensive evaluations for 28 leading VLMs. The experimental results show that the most advanced model like Gemini or GPT-4o still struggle to determine whether the question is answerable or not. This finding calls for further research and development of more reliable VLMs when handling unanswerable visual questions.

The main strength of this work is that the authors proposed a new benchmark to evaluate the trustworthness of VLMs. Through extensive evaluations for various VLMs, the authors showed that the top performed models still lack the ability to distinguish unanswerable questions from answerable ones. Clearly, this study conveys some new insights regarding the incapability of VLMs to cope answerable and unanswerable questions. Nevertheless, all reviewers pointed out that the proposed benchmark is heuristic and prone to unintendend bias introduced by the method used to synthesize unanswerable questions. This as a result will render biased evaluation results on the VLMs. With this, the ACs also agree and have a concern about the rigorousness of the proposed benchmark, given that all samples are designed heuristically and the total number of samples is also very limited. Furthermore, the ACs agreed with the reviewer gyxQ that the definition of answerable and unanswerable questions is a bit vague. For example, the question at Fig. 1 top right, "Is the output of the code ‘Hello’?" is considered as unanswerable question. However, this is more like a question that the model cannot answer deterministically. In this sense, the ACs suggest the authors make the definition of different types of questions more clear, so that the evaluations on top of this become more rigorous and indicative to the real capability of VLMs.

Based on the reviews and discussions durring rebuttal session, the ACs think this work still has some clear drawbacks which could be addressed by the authors in the revision. However, at this point, it does not pass the bar of this venue.

**Additional Comments On Reviewer Discussion:**

Some of the concerns raised by reviewers are addressed by the authors. However, there are still some major concerns regarding the solidness of the proposed benchmarks, and the way of curating unanswerable questions used by the authors. The ACs believe studing the trustworthness of LLMs and VLMs is of important topic and suggest the authors to take the comments into account the further polish the work.

---

### Decision · Program_Chairs · 2025-01-22

Reject